# Efficacy of Adductor Canal Block on Medial Knee Pain in Patients with Knee Osteoarthritis: A Randomized Single-Blind Placebo-Controlled Study

**DOI:** 10.3390/ijerph192215419

**Published:** 2022-11-21

**Authors:** Ki-Yong Kim, Yool-Gang Huh, Sang Hyeok Ma, Jong Hyeon Yoon, Kil-Yong Jeong, Do Young Park, Seung-Hyun Yoon

**Affiliations:** 1Department of Physical Medicine and Rehabilitation, Ajou University Medical Center, Suwon 16499, Republic of Korea; 2Department of Orthopedic Surgery, Ajou University Medical Center, Suwon 16499, Republic of Korea

**Keywords:** nerve block, osteoarthritis, knee joint, pain management, anesthetics

## Abstract

Background: This study aimed to confirm the efficacy of ultrasound-guided adductor canal block (ACB) as a treatment option for medial knee pain caused by knee osteoarthritis (KOA). Methods: In total, 31 participants with medial knee pain due to KOA were randomized to either the ACB (ultrasound-guided ACB, *n* = 15) or placebo group (1 mL of 1% lidocaine, *n* = 16). The primary outcome was a numerical rating scale (NRS) for knee pain intensity comparing before and 4 weeks after injection. The secondary outcomes were the Western Ontario and McMaster Universities Osteoarthritis Index (WOMAC), average daily number of analgesics consumed, average daily opioid consumption, and Timed Up and Go (TUG) test results before and 4 weeks after injection. Results: Participants’ baseline characteristics were not significantly different between the groups, except for age. At 4 weeks post-injection, the NRS score in the ACB group significantly improved compared to that in the placebo group (*p* = 0.009). However, the WOMAC, average daily number of analgesics consumed, average daily opioid consumption, and TUG test results did not show significant differences. Conclusion: ACB can be an effective treatment for reducing medial knee pain in patients with KOA.

## 1. Introduction

Knee osteoarthritis (KOA) is a common condition in the middle-aged and older population [1], and the most common symptom in patients with KOA is pain [2]. Pain is especially common in the medial compartment of the knee [3,4]. To reduce knee pain, pharmacological treatment is performed first. In addition, exercise, intra-articular injection, and surgery are used as non-pharmacological treatment options [5]. The saphenous nerve originates from the femoral nerve and is responsible for the sensation of the medial skin of the knee. Saphenous nerve block has been used to anesthetize the surgical site during lower extremity surgeries. As a means of achieving a higher anesthesia success rate, ultrasonography has been used to block the saphenous nerve inside the adductor canal, which is called ultrasound-guided adductor canal block (ACB) [6]. In addition to preoperative anesthetic purposes, it has been proven that ACB can reduce postoperative pain after total knee arthroplasty, meniscectomy, ligament reconstruction, osteotomies, and many other knee surgeries [7,8,9,10,11,12]. Considering previous studies that showed that ACB reduces pain in the medial compartment of the knee after the aforementioned knee surgeries, it can be assumed that it would have an effect of reducing pain in patients with KOA who complain of pain especially in the medial compartment of the knee. One retrospective study has suggested the efficacy of relieving pain when ACB was performed in patients with KOA [13], but no randomized controlled studies have proven this yet. 

If ACB is effective in controlling pain in patients with KOA, it may also have the advantage of improving patient function or reducing the use of oral analgesics, especially opioid consumption. The purpose of this study was to demonstrate the hypothesis that ACB is more effective in improving medial knee pain in patients with KOA than a placebo through a randomized, single-blind, placebo-controlled study.

## 2. Materials and Methods

### 2.1. Study Design and Population

This study was designed as a randomized, single-blind, placebo-controlled, single-center clinical trial conducted at a tertiary medical center in Suwon, Republic of Korea. Participants with KOA who visited the Department of Physical Medicine and Rehabilitation Outpatient Clinics were recruited from August 2018 to April 2022. Participants were identified only by number and not by name or initials. The inclusion criteria were as follows: (1) medial knee pain for at least 3 months and diagnosis of osteoarthritis by physical examination, radiography, and laboratory test results to exclude other conditions; (2) at least 45 years or older; and (3) Kellgren–Lawrence grading scale score of 2–4 for KOA. Exclusion criteria were (1) other causes of knee pain, such as rheumatoid diseases, infections, or fractures, and (2) prior surgery of the knee. Participants were allocated to either the ACB or placebo group via a block randomization method at a ratio of 1:1 with a block size of four. For group allocation, a computerized random number generator and table prepared by an investigator with no clinical involvement in the trial were used. The allocation procedure was concealed from the researchers. Participants were not informed of the group to which they belonged until the end of the final follow-up. Baseline characteristics, including age, sex, height, and weight, were recorded, and the location and duration of knee pain were surveyed. In addition, the type and number of doses of analgesics were checked using participants’ medical records or self-reports. To evaluate the effectiveness of the injection, participants visited the outpatient clinic 4 weeks after the injection.

### 2.2. Intervention

In both groups, the participants were placed in the supine position. Subsequently, the lead author (S.-H.Y.) placed a 10–13-megahertz linear ultrasound transducer (Logiq P6, GE Healthcare, Buckinghamshire, UK) on the participant’s medial mid-thigh. The saphenous nerve and femoral artery in the adductor canal were identified between the vastus medialis and adductor longus/magnus and below the sartorius muscle in the short-axis view. In the ACB group, a single-shot ACB was performed on the right, left, or both sides depending on the participant’s pain area. The lead author administered 10 mL of 1% lidocaine with a 23-gauge, 6 cm long needle under ultrasound guidance. In the placebo group, after identifying the sartorius muscle using ultrasonography, 1 mL of 1% lidocaine was injected into the sartorius muscle.

### 2.3. Outcome Measurements

The primary outcome was a numerical rating scale (NRS) score for knee pain intensity comparing before and 4 weeks after injection. The secondary outcomes were the Western Ontario and McMaster Universities Osteoarthritis Index (WOMAC), average daily number of analgesics consumed, average daily opioid consumption, and Timed Up and Go (TUG) test results before and 4 weeks after injection. The WOMAC is a validated self-reported questionnaire widely used to assess pain, stiffness, and physical functioning in patients with KOA [14,15,16]. It consists of 24 questions, each of which is scored from 0 to 4, meaning none, mild, moderate, severe, and extreme in ascending order, respectively. The scores were calculated for each dimension based on the sum of the scores. Each dimension’s score was added to calculate the WOMAC total score. We used the WOMAC translated into Korean, and it has also been validated in terms of reliability, validity, and responsiveness [17]. The TUG test is one of the easy, sensitive, and specific tests to assess patients’ functional abilities [18,19]. To conduct the test, participants were instructed to rise from a chair and walk at an appropriate and safe speed to a marker 3 m away, turn, return to the starting point, and then sit down again. The researcher measured the time from when participants were asked to start the test to when they were seated again. The average daily number of analgesics consumed was determined from the mean number of analgesics taken by the participants per day during the previous week. Examples of analgesics included acetaminophen, non-steroidal anti-inflammatory drugs, antispasmodics, and anticonvulsants. The average daily opioid consumption was the average amount of opioids consumed by participants per day during the previous week, which was converted into morphine equivalents. Examples of opioids included tramadol, oxycodone, codeine, and buprenorphine.

### 2.4. Statistical Analysis

In order to calculate the sample size, from the retrospective study described above, we used the mean and standard deviation of the visual analog scale of the knee pain level for the ACB and placebo groups at month 1 as reference values for the power analysis in this study [13]. With a power of 60%, a significance of 5%, and an effect size d of 0.85, 15 participants per group were required. Considering the number of potential dropouts, 32 participants were enrolled.

Since the group size was small and generally not normally distributed, the continuous variables were analyzed using the Mann–Whitney U test, except the WOMAC, which was normally distributed and analyzed using the independent t-test. Categorical variables were analyzed using the Fisher exact test because >20% of the expected frequencies had a value <5. To compare the differences before and after the intervention, the Wilcoxon signed-rank test was used, except for the WOMAC and TUG test results in the placebo group, which were normally distributed and analyzed using the paired t-test. For the WOMAC subscale analysis, the differences in each subscale before and after injection were compared. The pain and physical function subscales were analyzed using the Mann–Whitney U test, and the stiffness subscale was analyzed using an independent *t*-test. *p*-values < 0.05 were considered statistically significant. SPSS Statistics 25 (IBM Corp., Armonk, NY, USA) was used to perform all statistical analyses.

## 3. Results

A total of 32 participants were enrolled in this study, and 16 participants were randomly assigned to each group. Of these, one participant in the ACB group dropped out because he had been admitted to another clinic for a reason unrelated to KOA. Finally, 31 participants were evaluated for pain and functionality (Figure 1). The baseline characteristics of the two groups were not significantly different except for age (Table 1). As one participant in the ACB group was unable to walk because of pain, the TUG test could not be performed. For this reason, the data of that participant were excluded from the analysis of the TUG test results. Four weeks after the injection, NRS scores for knee pain and the WOMAC significantly improved only in the ACB group compared to the initial assessment from the within-group comparison. At the 4-week follow-up, the NRS score for knee pain in the ACB group was significantly lower than that in the placebo group based on the between-group comparison (Table 2). In the WOMAC subscale analysis, the ACB group made greater improvements than the placebo group only in the pain subscale (Table 3). No adverse events such as infection, hematoma, or dizziness occurred during the study.

## 4. Discussion

This study showed that ACB was effective in reducing medial knee pain in patients with KOA as the NRS score significantly improved in the ACB group compared to the placebo group, and the difference in WOMAC pain subscale scores between before and after injection in the ACB group was significantly different from that in the placebo group. In the ACB group, the average value of the NRS score difference before and after injection (2.1) was greater than the minimum detectable change in NRS score for pain in patients with KOA (1.33) [20]. Additionally, a previous study investigating minimal clinically important changes in chronic musculoskeletal pain suggested that one can call an improvement in NRS score of above two points “much better” [21]. Thus, it could be said that the pain reduction efficacy of ACB was significantly meaningful. However, indicators representing physical function, analgesic consumption, and opioid consumption were not improved by ACB.

From the apex of the femoral triangle to the adductor hiatus, the aponeurotic tunnel is called the adductor canal. It encompasses the superficial femoral artery, superficial femoral vein, and branches of the femoral nerve, e.g., the saphenous nerve and nerve to the vastus medialis. Additionally, it may contain the medial femoral cutaneous nerve and anterior cutaneous branch of the obturator nerve depending on the individual’s anatomical variation [22]. Previously, only the saphenous nerve block was considered to play an important role in ACB, but now, the nerve to the vastus medialis is also known to play a major role because it is believed to innervate not only the vastus medialis muscle but also the sensory of the joint capsule and medial retinaculum [23]. There is evidence that the injectate in the adductor canal can spread more proximally than the femoral triangle through vertical spread [24]. Therefore, it has been pointed out that injectate may spread out of the femoral triangle and become an indirect femoral nerve block. However, in several clinical trials, quadriceps weakness due to vertical spread was shown to be insignificant, and ACB is associated with less quadriceps weakness [25,26,27]. This may be because motor nerves to the quadriceps muscles branch out just below the inguinal canal, where the local anesthetic generally does not spread vertically. The only motor nerve that can be affected by mid-thigh ACB is the nerve to the vastus medialis. Therefore, ACB has similar pain reduction as a femoral nerve block but is associated with less quadriceps weakness, and it is linked with earlier ambulation or better functional outcomes [26,27]. Accordingly, motor weakness of the lower limbs was not reported in this study.

The advantage of ACB is its efficacy in reducing pain, even in patients with late-stage KOA who have severe knee pain and very poor physical dysfunction. The average NRS score of all participants in this study was 7.2 ± 1.4, and 26 of 31 (83.9%) participants had an NRS score ≥6, which means that the participants had moderate to severe knee pain [28]. Additionally, 29 of 31 (93.5%) participants had a Kellgren–Lawrence grade of 3 or 4, which also indicates radiographically moderate to severe KOA. Of the 31 participants, 29 had moderate to severe physical dysfunction with a WOMAC total score ≥36 [29]. One of the contraindications of total knee arthroplasty is poor physical functioning [30]. Judging from the fact that the average WOMAC of all participants was 54.13 ± 13.16, their physical function would have been very poor and surgical treatment could have been difficult. In fact, some patients’ medical records showed that total knee replacement surgery was not possible or that the benefits were lower than the risks. In this context, ACB can provide pain relief even in patients with KOA severe enough to make surgical treatment difficult. Second, the results indicated that the efficacy in terms of pain relief persisted for at least 4 weeks with a single injection. Since most previous studies related to ACB focused on postoperative pain control, the effect was often followed up for only 24–72 h. However, in this study, follow-up was performed after 4 weeks, and participants’ pain was proved to be still reduced; thus, it can be seen that the pain-reducing efficacy of ACB lasted for at least 4 weeks. Although the typical elimination half-life of lidocaine is known to be about 3 h [31], one of the reasons ACB could reduce pain up to the 4-week follow-up in this study can be related to attenuation of central sensitization. Central sensitization can be defined as an amplification of neural signaling within the CNS that elicits pain hypersensitivity and hyper-responsiveness to nociceptive stimuli [32,33]. As repeated nociceptive inputs trigger, reinforce, and maintain central nociceptive circuits [32], treatments to reduce peripheral nociception can potentially attenuate central sensitization [34]. A number of studies have shown that central sensitization is also involved in the development and maintenance of chronic pain in KOA [35,36,37]. Furthermore, there is evidence that reducing pain in KOA patients has desensitized nociceptive circuits. For example, one study revealed that central sensitization in patients with KOA was decreased after undergoing joint replacement [38]. Therefore, even if it is not possible to alternate disease progression and inflammatory consequences through ACB, the pain can be temporarily removed to achieve a longer effect of pain relief for KOA patients through a pain desensitization and central remodeling process. Third, hyperglycemia, which is a potential risk of corticosteroid injection [39], would be minimal because corticosteroids were not mixed with the injection. In other words, it can be safely and repeatedly administered to patients with diabetes who are at risk of high blood sugar levels.

Although ACB seemed to significantly reduce pain, as can be seen from the NRS scores and WOMAC pain subscale results, there was no significant effect on improving physical function, as shown by the TUG test or WOMAC physical function subscale results. This may be because pain reduction did not immediately lead to improvement in physical function. The average age of all participants was 71.4 ± 1.9 years, and symptom duration was 8.8 ± 1.0 years. Most of them had late-stage KOA with a Kellgren–Lawrence grade of 3 or 4 and moderate to severe physical dysfunction. As the participants were elderly with very poor physical function for long periods of time, it may be assumed that their physical function was deconditioned enough already, not just because of the pain itself, and even if the pain was reduced temporarily, it may have been difficult to recover physical function within a short period of 4 weeks. If the participants were relatively younger and had shorter symptom durations, lower severity, and longer follow-up periods, their physical function might have improved.

This study has a few limitations. First, the power of the conclusion is relatively low at about 60%. Second, there was a significant difference in the average age of the two groups. However, there was no significant difference between the two groups for other important variables such as NRS score, physical function indicators, and doses of painkillers, which are primary or secondary outcomes. Third, the sample size was small, which might explain why there were no significant differences in other results such as physical function. Finally, the follow-up period was short, and the number of follow-ups was small. In this study, only one follow-up was performed 4 weeks after injection. If further studies involve longer periods with a larger number of participants, mid- to long-term changes could be confirmed.

## 5. Conclusions

This randomized, single-blind, placebo-controlled study investigated the efficacy of ACB in patients with KOA. It is meaningful in that this study reveals the efficacy of ACB which was suggested in previous studies. ACB may be an effective treatment for reducing medial knee pain in patients with KOA. In addition, it could be an alternative treatment option for patients with KOA who have difficulty with pharmacological treatments due to gastrointestinal side effects or with surgical treatments due to comorbid medical conditions.

## Figures and Tables

**Figure 1 ijerph-19-15419-f001:**
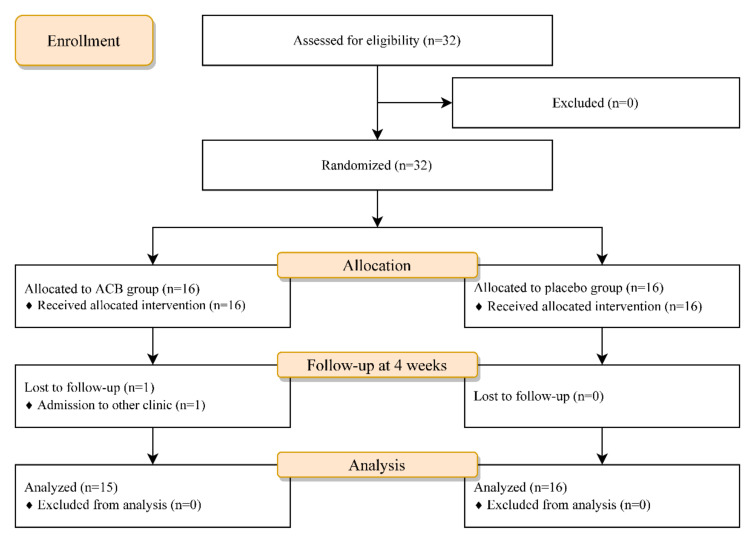
CONSORT flow diagram showing the progress of participants. ACB = adductor canal block.

**Table 1 ijerph-19-15419-t001:** Participants’ baseline characteristics.

	ACB Group (*n* = 15)	Placebo Group (*n* = 16)	*p*-Value
Age, y	76.8 ± 8.9	66.3 ± 9.9	0.008 *
Sex, men:women, *n*	3:12	4:12	0.539
Duration of symptoms, y	8.5 ± 5.2	9.0 ± 6.3	0.892
Body mass index, kg/m^2^	25.7 ± 5.6	26.9 ± 6.4	0.800
Kellgren–Lawrence grade, 2:3:4, *n*	0:4:11	2:4:10	0.595
Site, right:left:both, *n*	3:1:11	3:0:13	0.820
NRS score	7.1 ± 1.6	7.3 ± 1.2	0.800
WOMAC	54.6 ± 15.3	53.7 ± 11.3	0.851
Average daily no. of analgesics consumed, *n*	2.4 ± 2.2	2.8 ± 2.5	0.740
Average daily opioid consumption, mg	8.2 ± 18.4	8.4 ± 29.8	0.520
Timed Up and Go test result, s	29.8 ± 20.2 ^†^	17.8 ± 10.3	0.058

Values are expressed as mean ± standard deviation except sex, the Kellgren–Lawrence grade, and site, which are expressed as n. ACB = adductor canal block; NRS = numerical rating scale; WOMAC = Western Ontario and McMaster Universities Arthritis Index; no. = number. * Statistically significant. ^†^
*n* = 14 because one participant was not tested.

**Table 2 ijerph-19-15419-t002:** Changes in the outcome measurements before and after injection.

	ACB Group (*n* = 15)	*p*-Value, Within-Group Comparison	Placebo Group (*n* = 16)	*p*-Value, Within-Group Comparison	*p*-Value, Between-Group Comparison
NRS score		0.002 *		0.253	
Pre-injection	7.1 ± 1.6		7.3 ± 1.2		0.800
Post-injection	4.9 ± 2.4		6.9 ± 1.5		0.009 *
WOMAC		0.002 *		0.062	
Pre-injection	54.6 ± 15.3		53.7 ± 11.3		0.851
Post-injection	44.7 ± 19.1		49.4 ± 16.4		0.467
Average daily no. of analgesics consumed, *n*		0.414		0.450	
Pre-injection	2.4 ± 2.2		2.8 ± 2.5		0.740
Post-injection	2.1 ± 2.3		3.0 ± 3.0		0.520
Average daily opioid consumption, mg		0.157		0.715	
Pre-injection	8.2 ± 18.4		8.4 ± 29.8		0.520
Post-injection	7.0 ± 18.2		6.2 ± 17.3		0.892
Timed Up and Go test result, s		0.064		0.898	
Pre-injection	29.8 ± 20.2 ^†^		17.1 ± 10.3		0.058
Post-injection	34.9 ± 47.8 ^†^		17.7 ± 10.1		0.193

Values are expressed as mean ± standard deviation. ACB = adductor canal block; NRS = numerical rating scale; WOMAC = Western Ontario and McMaster Universities Arthritis Index; no. = number. * Statistically significant. ^†^
*n* = 14 because one participant was not tested.

**Table 3 ijerph-19-15419-t003:** WOMAC subscale analysis.

	ACB Group (*n* = 15)	Placebo Group (*n* = 16)	*p*-Value
ΔWOMAC_pain	2.8 ± 2.5	0.1 ± 2.7	0.012 *
ΔWOMAC_stiffness	0.3 ± 1.2	0.8 ± 1.6	0.369
ΔWOMAC_physical function	6.8 ± 9.2	3.4 ± 7.2	0.401

Values are expressed as mean ± standard deviation. Δ is the difference between the pre- and post-injection values. ACB = adductor canal block; WOMAC = Western Ontario and McMaster Universities Arthritis Index. * Statistically significant.

## Data Availability

The data that support the findings of this study are available from the corresponding author upon reasonable request. The data are not publicly available due to privacy restrictions.

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
