# Peer review of "Efficacy of Adductor Canal Block on Medial Knee Pain in Patients with Knee Osteoarthritis: A Randomized Single-Blind Placebo-Controlled Study"

_ijerph, 2022, doi:10.3390/ijerph192215419_

Round 1

Reviewer 1 Report

Thank you very much for giving me the opportunity to review your very valuable paper. I read it with great interest.

However, why did the effects of ACB last for 4 weeks? Please share your thoughts on that in the discussion. Can we conclude that the ACB did reduce pain compared to placebo, but that the degree of pain reduction was very small?

Also related to the above, is there any possibility of significantly improving WOMAC and TUG by increasing the degree of pain relief by, for example, repeating ACB many times, once a week? Ideas on how to improve the method and frequency of such ACBs are also in the discussion.

If you could add a note about the above, this article would be more beneficial to our readers.

Reviewer 2 Report

This is a well-made study, and I can see and applaud the effort you made writing this paper and submitting it to IJERPH (ISSN 1660-4601). Knee osteoarthritis is a common disease in middle-aged and older people. Pain is the main complaint and a major cause of chronic disability, occurring most often in medial knees. This study would seem to evaluate the effectiveness of pain relief and functional improvement in KOA patients treated with CBA. The review process aims to improve your manuscript, even when it might not be suitable for publication here. Please meticulously consider the comments and feedback given in the following. Do not take it personally, as the only aim is to improve your scientific manuscript. Generally, the topic is of interest to the IJERPH readership. However, it has some problems, which are described below. Furthermore, please consider the comments below and make the proper adjustments.

Abstract

Lines 22-23:

Rephrase the final sentence of the conclusion better. You talk about alternatives versus what? Perhaps better to talk about a valid therapeutic approach without mentioning other therapies that you may have to relate to eventually.

Introduction

Lines 37-38:

This procedure is proper in TKA and meniscectomies and cruciate ligament reconstruction and osteotomy procedures.

I suggest you add this part, "total knee arthroplasty, meniscectomy, ligament reconstruction and osteotomies, and many other knee surgeries..." in support of the procedures I suggest you add the following citations: doi: 10.1016/j.knee.2020.12.017, doi: 10.1007/s00590-022-03419-4., doi: 10.21037/aoj-22-19.

Lines 38-39:

Here you refer to previous studies but report only one reference. Please include the other studies you refer to (insert references).

Lines 47-48:

Please give your exact hypothesis in detail here. The reader also needs to understand why your study is clinically relevant. What makes your study more appropriate than others already published? What is the news? What makes your study necessary? What is different? What is comparable? You need to state what is original in your research. What makes it worth publishing?

Materials and Methods:

The patient recruitment period is unclear. Please include it.

Has an ethics committee approved the study? If it has, please add the protocol number.

Line 88:

Add references pointing to studies that have used Womac to confirm the strength of the score used. I suggest you add: doi: 10.1007/s00167-018-4879-5, doi: 10.1016/j.jor.2022.06.014.

Line 95:

"m" stands for meter? Please specify

Line 106:

Why do you refer to a retrospective study if yours is a randomized, single-blind, placebo-controlled, single-center clinical trial? Please correct this part.

Results:

Okay

Tables: 

Your measurement methods must be given in detail. Measurement accuracy is necessary to report. Please make sure that your results are given with the same accuracy as the methods. If your methods allow one decimal, the result should also be reported with one decimal. Information about measurement accuracy is essential.

Discussion

Line 160-161:

Please always speak in the third person. Avoid "our".

This first sentence should be in the introduction, not the discussion.

Lines 220-224:

Limitations of your study must be added and discussed in detail somewhere close to the end of the Discussion section. This is always an important part of every manuscript and is something that will lead to new scientific studies in the future.

For example, you must add that you have a low power analysis of 60%. It is considered optimal when it is at 80%. A significant limitation is the age difference between the two groups, which must be considered and arguments. Also, you have to add that further studies with greater follow-up and larger numbers of patients would be necessary to confirm what you found with your research.

In the end, please mention the clinical relevance of your work. How can this work be helpful in the day-to-day clinical work?

References

Please add the references suggested above.

Reviewer 3 Report

POSITIVE POINTS:

The article is important.  This work matter to physical therapist, clinicians, researchers, and educators in physiotherapy and medicine and it help your readers to make better decisions. This manuscript add enough to existing knowledge. This work add enough to what is already in the published literature. We can read well this work and make sense. It has a clear message.

o The research question is clearly defined and appropriately answered.

o The design of study is appropriate and adequate to answer the research question.

o Participants are adequately described, their conditions defined, inclusion and exclusion criteria are described.

o Methods are adequately described.

-          Outcomes measures are clear and are clinically important.

-          The allocation to treatments was randomised. The method of randomisation was adequately described. The allocation of patients to the different arms of the trial was arranged so that no one could predict which treatment the patient would receive.

o The results answer the research question, and are credible and well presented.

o Interpretation and conclusions warranted by and sufficiently derived from the data. The authors discusses in the light of previous evidence and the message is clear.

o The references are relevant and up to date.

o Abstract/summary/key words/what this paper adds reflect accurately what the paper says.

IMPROVEMENT POINTS:

- The approval of the clinical trial by an ethics committee is not mentioned.

- It is not mentioned if the protocol of this research was submitted to a clinical trial registry.
